# Higher Serum Phosphorus Is Not an Independent Risk Factor of Mortality in Heart Failure with Reduced Ejection Fraction

**DOI:** 10.3390/nu13114004

**Published:** 2021-11-10

**Authors:** Partyka Robert, Mroczek Alina, Duda Sylwia, Malinowska-Borowska Jolanta, Buczkowska Marta, Głogowska-Gruszka Anna, Niedziela Jacek, Hudzik Bartosz, Gąsior Mariusz, Rozentryt Piotr

**Affiliations:** 1Clinical Division of Anesthesiology and Intensive Therapy of the Department of Anesthesiology, Intensive Treatment and Emergency Medicine, Medical University of Silesia, 41-800 Zabrze, Poland; robertpartyka@op.pl; 2Department of Toxicology and Health Protection, Faculty of Health Sciences in Bytom, Medical University of Silesia in Katowice, 41-902 Bytom, Poland; amroczek@sum.edu.pl (M.A.); sduda@sum.edu.pl (D.S.); jmalinowska@sum.edu.pl (M.-B.J.); mbuczkowska@sum.edu.pl (B.M.); aglogowska@sum.edu.pl (G.-G.A.); 3Third Department of Cardiology, SMDZ in Zabrze, Medical University of Silesia in Katowice, Silesian Centre for Heart Disease, 41-800 Zabrze, Poland; jacek.niedziela@gmail.com (N.J.); bhudzik@sum.edu.pl (H.B.); m.gasior@sccs.pl (G.M.); 4Department of Cardiovascular Disease Prevention, Faculty of Health Sciences in Bytom, Medical University of Silesia in Katowice, 41-902 Bytom, Poland

**Keywords:** serum phosphorus, chronic heart failure, prognosis, all-cause mortality

## Abstract

Higher serum phosphorus has detrimental health effects. Even high-normal rage sP is associated with worse outcomes. The relationship of serum phosphorus with prognostic markers in heart failure remains unclear. We investigated the association of serum phosphorus with heart failure prognostic factors and risk of mortality related to serum phosphorus. In 1029 stable heart failure patients, we investigated the distribution of markers of more advanced heart failure stage across quintiles of serum phosphorus and estimated the relative risk of mortality in comparison to reference. Higher serum phosphorus levels sP were associated with markers of a worse outcome. The best survival was observed in low-normal serum levels. The unadjusted hazard ratio for mortality increased toward higher phosphorus quintiles but not to lower levels of sP. The correction for age, sex, BMI, percent weight loss, inflammation, kidney function, and LVEF did not modify the risk profile substantially. The adjustment for NYHA, natriuretic peptides, serum sodium, and treatment characteristics broke down the risk relationship completely. A higher serum phosphorus is associated with markers of a more risky profile of heart failure. Elevated serum levels of phosphorus sP does not provide independent prognostic information beyond the strongest markers of the severity of the syndrome. The potential involvement of higher serum phosphorus as a mediator in the pathophysiology of heart failure warrants further study.

## 1. Introduction

Phosphorus is a key nutrient for living organisms. The element is important for energy production, proper function of nucleic acids and enzymes, cell proliferation and signaling, and as a component of the skeleton [1,2]. The exit from physiological range leads to detrimental health consequences. For example, low serum phosphorus (sP) is a known risk factor for development of heart failure (HF) and aggravates clinical profile of existing HF [3]; in patients with septic shock, reduced sP increases their morbidity and mortality [4]. Elevated sP is known to initiate cardiovascular remodeling and clinical events in renal patients [5,6]. However, in non-renal populations, even high-normal sP has been associated with accelerated atherosclerosis [7] and excessive morbidity and mortality [8,9,10].

HF has reached an epidemic proportion worldwide [11]. It is characterized by multiple endocrine abnormalities, malnutrition, and altered kidney function [12], all of which are associated with changes of phosphorus metabolism and alteration of serum levels.

Although numerous studies showed the association of high normal sP and worse outcome in general population and various diseases [8,9], very little is known about such an association among HF patients [13,14,15]. It is unclear why sP is elevated in high proportion among HF patients even with preserved kidney function and whether these elevated levels are associated with a poor outcome independently of known markers of HF severity.

We attempted to determine the association of sP with markers of HF severity as well as with intensity of HF therapy. Finally, we investigated whether sP was associated with all-cause mortality independently from these variables.

## 2. Methods

### 2.1. Study Group

We analyzed patients included in the Prospective Registry of Heart Failure conducted in our department since 2003. We selected patients with reduced left ventricle ejection fraction (LVEF ≤ 40%), diagnosed according to the current European Society of Cardiology criteria as having HF, with age > 18 years, and duration of signs and symptoms of more than 6 months [16]. They had received optimal therapy for at least 1 month. All inclusion patients had no signs or symptoms of volume overload. Patients who received glucocorticosteroids, bisphosphonates, vitamin D formulations, or calcium- or phosphorus-containing salts and those who had an active infection, liver disease with enzymes exceeding 4 times the upper reference limit, active bleeding, known neoplasm or granulomatous disease, and patients with a history of any surgery reducing gut absorption capacity were excluded from the study.

Study criteria were fulfilled in 1029 registry participants, and their data were included in the final analysis.

Based on the medical history, we established the date of HF onset with a one-month precision and calculated the duration of HF symptoms. Using medical records, we determined the highest body weight within a year before the development of HF, taken as the pre-HF body weight. When multiple measurements were available in individual patients, the mean value was taken as pre-HF body weight.

Comorbidities, such as hypertension, diabetes mellitus, and hypercholesterolemia, were recognized based on medical history, current therapy, or laboratory values of respective variables. A history of smoking was defined as current or previous use of tobacco products.

Blood samples were drawn in a standardized fashion, between 8 and 10 am, after at least 8 h of fasting and after 30 min of rest in a supine position in a quiet, environmentally controlled room. Blood was immediately centrifuged at 4 °C and stored at –75 °C for further analysis. All procedures were in accordance with Helsinki Declaration, and the protocol was approved by the Bioethics Committee of the Medical University of Silesia. All patients expressed their informed, written consent.

### 2.2. Measurements

Body mass and height were measured on the day of blood sampling (index date) using a certified scale (B150L, Redwag, Zawiercie, Poland). Body mass indexes (BMI) were calculated for pre-HF and index date weight and are shown as pre-HF BMI and indexBMI, respectively. We calculated weight loss during HF based on formula shown below:Weight loss [%] = 100 × [(pre-HF BMI − indexBMI)/pre-HF BMI]

We used Sonos-5000 Hewlett-Packard Ultrasound Scanner (Hewlett-Packard, Andover, MA, USA) to measure LVEF from the apical four-chamber view and calculated it with the formulation:LVEF = [(end-diastolic volume − end-systolic volume)/end-diastolic volume] × 100

Commercially available reagents (Roche Diagnostics, Basel, Switzerland) allowed measurements of serum creatinine, N-terminal brain-type natriuretic pro peptide (NTproBNP), high sensitivity C-reactive protein (hsCRP), serum sodium, albumin, phosphorus, and calcium. We calculated GFR using the known MDRD formulation:eGFRMDRD = 186 × plasma creatinine [mg/dL] − 1.154 × age [years] − 0.203 × 0.742 (if female).

For serum albumin <40 g/L, we corrected calcium using the formulation:Corrected calcium [mmol/L] = total calcium + 0.02 × (40 − plasma albumin [g/L]).

We did not measure ionized calcium.

### 2.3. Statistics

Continuous variables are presented as means and standard deviations, categorical as percentages. For parameters with a skewed distribution, medians and interquartile ranges (IQR) are given. To correct non-normal distribution of variables in calculations, we used log_10_ transformed values (in tables shown before transformation). The study cohort was divided into sP quintiles, and a comparison of parameters between quintiles was carried out using Kruskal–Wallis or chi-square tests, where appropriate. For each quintile, we calculated the crude all-cause mortality rate based on 18-month follow-up. Data on mortality were obtained from the National Database on Inhabitants. We constructed Kaplan–Meier cumulative survival curves and compared the survival of phosphorus quintiles using log-rank test. Cox proportional hazard model was used to estimate the risk of death by taking quintile with the lowest mortality as a reference. In each quintile, we calculated relative hazard (HR) and 95% confidence intervals (CI) in unadjusted, minimally adjusted, and fully adjusted models. The adjustment parameters were based on the significant differences between the quintiles and on clinical reasoning. Finally, we applied cubic spline modelling to estimate the association between sP and the risk of all-cause mortality in either unadjusted or adjusted models. The significance level was set at 0.05 (two-tailed), and all calculations were performed using packages of “Statistica” v.10.0 and “R” version 3.1.0 (2014).

## 3. Results

### 3.1. Clinical and Laboratory Characteristics

In groups with higher sP, there were more women. Patients were younger, and they lost more weight in a shorter duration of HF, resulting in lower indexBMI. Their LVEF was lower and their NYHA class higher. They also had worse kidney function and higher NTproBNP, hsCRP, and calcium but lower serum sodium. Patients with higher sP received lower doses of ACEI/ARB but higher aldosterone antagonists and loop diuretics. They were also treated more frequently with digoxin. The comorbidity profile did not differ between the quintiles of SP. Detailed characteristics of patients included in the study and comparison of patients in quintiles of sP are shown in Table 1. In general, patients in ascending quintiles of sP had a more advanced clinical and laboratory profile of HF, and their treatment was less intense.

Among patients in the bottom quintile of sP, hypophosphatemia with levels below 0.81 mmol/L was observed in 57 (28.3%) patients, while in the top quintile of sP, hyperphosphatemia with levels exceeding 1.40 mmol/L was recognized in 108 (52.4%) patients.

The clinical and laboratory characteristics of the hypophosphatemic patients and the remaining low normal patients of quintile 1 were not different, with the exception of lower NTproBNP: 869 (972) pg/mL versus 1203 (1813) pg/mL, *p* < 0.001; and lower serum calcium: 2.31 ± 0.16 mmol/L versus 2.37 ± 0.14 mmol/L, *p* < 0.001 in patients with hypophosphatemia (not shown in Table 1).

Patients with hyperphosphatemia compared to high-normal patients in quintile 5 had more advanced NYHA class: (I < 1%, II—24%, III—55%, IV—21%) versus (I—1%, II—39%, III—50%, IV—10%), *p* = 0.009, higher serum calcium: 2.47 ± 0.16 versus 2.43 ± 0.18, *p* = 0.04, and lower eGFR_MDRD_: 64 (47) mL/min × 1.73 m^2^ versus 79 (42) mL/min × 1.73 m^2^, *p* = 0.008.

### 3.2. Unadjusted and Adjusted Risk of All-Cause Mortality

During 18 months of follow-up, 185 patients (18%) died. The crude mortality rate was the lowest in quintile 2 and increased in both extremes of sP. The difference in mortality between quintiles was significant (Table 1).

The cumulative probability of death represented by the Kaplan–Meier curves also showed significant differences between the sP quintiles (*p* = 0.02) (Figure 1).

The cumulative survival of patients with hypophosphatemia and patients with low-normal sP from quintile 1 was similar (log-rank *p* = 0.32) (Figure 2).

Mortality in patients with hyperphosphatemia and patients with high-normal phosphorus from quintile 5 was also comparable (log-rank *p* = 0.58) (Figure 3).

An unadjusted Cox analysis with quintile 2 as a reference showed increasing risk of mortality toward quintiles with higher sP (*p* = 0.03 for trend). Hazard ratio peaked in quintile 5, reaching a value of 2.15; (95% CI: 1.35–3.53, *p* = 0.003) (Table 2). The risk of death in a group with the lowest sP was not different from the reference. The cubic-spline modelling of an unadjusted relation between sP and mortality is presented in Figure 4.

By fitting serum calcium into the unadjusted model, we tried to disclose the possible interaction of serum calcium and phosphorus as risk factors for mortality. The adjustment has changed the risk marginally in the lowest sP; however, the risk of death was increasingly attenuated in each incremental quintile of sP. The greatest reduction in the hazard ratio was observed in the top quintile. Despite HR attenuation, the risk in patients in two upper quintiles of sP remained significantly higher compared to the reference (Table 2, model 1).

In the next step, we examined the influence of sex, age, BMI, weight loss, hsCRP, eGFR_MDRD_, and LVEF but not calcium on the association between sP and mortality. The analysis showed a slight reduction in risk attributable to higher sP. However, the HR of patients in the two upper quintiles of sP still remained significantly elevated (Table 2, model 2).

As variation in treatment and differences in functional status, serum sodium and NTproBNP could potentially be linked to a worse prognosis; the association between sP and mortality was further adjusted for these variables. When we added the use of ACEI/ARB, beta-blockers, loop diuretics, and digoxin as categorial variable and a percentage of recommended doses of ACEI/ARB, aldosterone antagonists, and the dose of loop diuretics as continuous variables as well as NYHA class, serum sodium, and NTproBNP levels, the association of mortality with increasing quintiles of sP was no longer significant (Table 2, model 3).

In Figure 5, the fully adjusted relation between sP and mortality is presented as a cubic spline model.

Our final analysis was designed to identify the risk attributable to sP taken as a continuous variable together with all potential confounders, such as age, sex, BMI, weight loss, hsCRP, eGFR_MDRD_, serum calcium, LVEF, serum sodium, NTproBNP, the use of beta-blockers and digoxin, loop diuretic dose, and the percentage of the recommended dose of ACEI/ARB and aldosterone antagonist administered to patients. In this final model, the only significant independent predictors of mortality were NYHA class (HR = 1.92; 95%CI: 1.42–2.59, *p* < 0.001), NTproBNP (1.08; 95%CI: 1.04–1.13, per 1000 pg/mL increase, *p* < 0.001), serum calcium (HR = 1.24; 95%CI: 1.02–1.51 per 0.2 mmol/L increase, *p* = 0.02), and loop diuretic dose (HR = 1.12; 95%CI: 1.02–1.24, per 40 mg of furosemide equivalent increase, *p* = 0.02). Serum phosphorus was not a significant predictor. The relationship between sP and mortality after full adjustment is shown on Figure 6.

As there was no difference in mortality between patients with hypophosphatemia and those with low-normal sP of quintile 1 in Kaplan–Meier analysis (Figure 2), we constructed Cox proportional hazard model to estimate the risk of hypophosphatemia after accounting for factors that were different between these subgroups. After adjusting for NTproBNP and serum calcium, the hazard ratio for mortality was HR = 0.78; 95%CI: 0.34–1.82, *p* = 0.58. Comparison of hypophosphatemic patients with all patients with normal sP did not significantly change the results.

Similar analysis was performed in patients with hyperphosphatemia as compared to those with high-normal sP from quintile 5 (Figure 3). Adjusting for differences in NYHA class, serum calcium, and eGFR_MDRD_ that exist between these subgroups in the Cox model did not reveal a higher risk in death of hyperphosphatemic patients with HR = 1.04, 95%CI: 0.58–1.88, *p* = 0.88. The inclusion of all patients with normal phosphorus compared to hyperphosphatemic patients did not modify the risk profile.

## 4. Discussion

The link between altered phosphorus metabolism and prognosis in HF has been debated [13,14,15,17,18]. There is a significant heterogeneity between studies with respect to patient clinical characteristics [13,14,17], kidney function [15,18], duration of follow-up [19,20], and the study outcome [15,17]. In addition, important confounders were taken into account only in the small amount of studies [15,18]. Some studies did not account for standard prognostic factors in HF [17] or only made minimal adjustments [18]. As a result, the conclusions of these studies are inconsistent.

Our study reports all-cause mortality in a relatively large cohort of optimally treated HF with reduced ejection fraction. We found the lowest death rates in a range of sP between 0.95 and 1.04 mmol/L. There was a gradual increase in mortality in the ascending quintiles of sP but not at the lowest levels. In the bottom and top quintiles of phosphorus, there were subgroups with hypophosphatemia and hyperphosphatemia, respectively. Apart from sP, in these subgroups, important HF severity markers were different in comparison to patients with normal sP. Despite these differences, the prognosis for these patients was similar.

Many markers of more severe HF aggregated in groups of higher sP, while therapy modifying prognosis was less intensive in these patients. Younger patients and women were more frequent among those with higher sP. This distribution probably represents the effect of splitting into quintiles of phosphorus because higher levels occur naturally in younger people and in women [19]. However, since these characteristics also affect the prognosis, they were included in the analysis as potential covariates.

In the multivariable analysis, we found a significant association between higher sP and all-cause mortality that was independent of age, sex, BMI, eGFR_MDRD_, inflammation, LVEF, and weight loss. However, when we considered differences in therapy, NYHA class, and neuroactivation markers, such as serum sodium and NTproBNP, this association disappeared.

In one investigation that included a population similar to ours, the association of sP with the composite of death and heart transplantation was independent of NYHA class and natriuretic peptides [14]. This discrepancy may be explained by differences in clinical outcome because in this study, up to 24% of the composite end point was driven by heart transplantation alone [14]. Another reason for the inconsistent result may come from the stratification by gender used in the study mentioned above but not in ours.

In a much smaller study by Plischke, with a twice-longer follow-up [15], sP significantly predicted the composite of death or cardiac hospitalization [15]. Again, difference in length of follow-up and study outcome may explain the observed discrepancy.

The key questions when studying the prognostic role of phosphorus in HF are the reasons for the elevation of phosphorus in more severe HF and why sP correlates with numerous markers of a poor prognosis.

Many authors argue that reduced kidney perfusion in HF alters the renal excretory potential, leading to phosphorus accumulation [20]. Poor heart function and impaired kidney filtration are both recognized as reasons for poor clinical outcomes in HF [21]. In this interpretation, higher sP would be secondary event in response to the underlying cardio-renal interaction.

Our data challenged this interpretation. Among patients with reduced eGFR_MDRD_ (<60 mL/min × 1.73 m^2^) and with preserved eGFR_MDRD_ (≥60 mL/min × 1.73 m^2^), there was a huge overlap of sP. Hypophosphatemia was present in patients with low kidney function, while hyperphosphatemia was seen in normal eGFR_MDRD_ (Figure 7).

Similar overlap was also observed when the entire cohort was divided by median sP (Figure 8).

Additionally, in multivariable analysis, the increase in the risk in the ascending quintiles of sP was independent of both LVEF and eGFR_MDRD_, suggesting the contribution of other mechanisms.

Among potential mechanisms, one may point to hypoxia and energetic insufficiency at the cellular level. Intracellular phosphorus transport is coupled with energy-consuming sodium pumping in the opposite direction, which gives a primary force to intracellular phosphorus transport [22]. Low serum sodium may inhibit intracellular transport, suggesting that sP elevation may be a secondary event. Low serum sodium and elevated natriuretic peptide both reflect disturbed neurohormonal regulation and, according to current understanding, represent central pathways in HF pathophysiology [23]. Previously, we showed that among the significant sP predictors in HF, the strength of associations between mortality and NTproBNP, serum sodium, and degree of catabolism was similar to that of eGFR_MDRD_ [24]. These observations suggest high complexity of processes underlying sP increase in more ill patients with HF. Based on this, we believe that the correlation between higher sP and markers of more severe HF may stem from common pathophysiological pathways, in which cardio-renal interaction is only one piece of the puzzle. Therefore, a higher sP may be a downstream event of the key pathophysiological pathways in HF.

This hypothesis was further supported by the observation of less intensive pharmacotherapy in patients with higher SP and also by the fact that the association between sP and mortality was broken down by accounting for differences in pharmacotherapy. At a more advanced stage of HF, there is a limited use of ACEI/ARBs due to intolerance but more frequent application and use of higher doses of loop diuretics and aldosterone antagonists [25]. We previously showed that in patients with less intensive treatment and worse clinical response to therapy, hyperphosphatemia occurred more frequently [26].

Another important question is whether, for whatever reason, increased sP poses an additional cardiovascular risk in HF beyond that related to known pathways.

Most animal experiments support the direct role of excessive sP on the cardiovascular system. They showed decreased vascular nitric oxide bioavailability, up-regulation of angiotensin converting enzyme expression, and induction of vascular calcifications in animals exposed to higher sP [27,28]. In a study of uremic rats, high phosphorus diet-induced hyperphosphatemia was shown as a direct stimulator of systemic inflammation, malnutrition with body wasting, and vascular calcification [29].

In our study, the association between mortality and a higher sP remained significant after adjustment for systemic inflammation, BMI, and degree of weight loss. This observation is in agreement with the animal experiment cited above and suggests that elevated sP might be, at least in part, an upstream event for inflammation and weight loss, all known risk factors for mortality in HF [30].

In in-vivo experiments of humans, endothelial dysfunction appeared after exposure to a diet rich in phosphorus, leading to elevation of sP [31]. Not only that, but cellular experiment clearly demonstrated induction by phosphorus of apoptosis within cardiomyocytes in culture [32]. Furthermore, calcification of vascular smooth muscle cells was recognized in patients with elevated phosphorus and calcium [33,34]. In a very recent study in humans with HF with reduced ejection fraction, the higher propensity for calcification as estimated by function T-50 test was associated with worse outcome in patients with ischemic etiology [35].

It is worth noting that when we account for different serum calcium levels in the multivariate analysis, we found a preferential reduction of risk in higher quintiles of phosphorus. When we compared the hazard ratios for mortality in unadjusted and calcium-only adjusted models, the risk reduction in the later model was 1% in quintile 1, while 3% and 15% in quintiles 4 and 5, respectively. This observation may suggest a complex interaction between calcium/phosphorus abnormalities and mortality, especially in higher sP groups. Indeed, a previous study showed a significant prediction of total mortality in HF by calcium phosphorus product and serum calcium but not by sP alone [13]. Our final result of multivariable prediction of mortality was in agreement with this observation. This analysis showed that apart from NTproBNP, NYHA class, and loop diuretic dose, serum calcium but not phosphorus remained significantly associated with the outcome.

In addition to the direct effect related to phosphorus, typical hormonal adaptations to phosphorus loading, such as higher parathyroid hormone, fibroblast growth factor 23, and lower 1.25-dihydroxy vitamin D [2], all were suggested to increase cardiovascular morbidity and mortality [18,36,37]. Unfortunately, serum levels of calciotropic and phosphaturic hormones were not available in our study, and we were unable to perform a more in-depth analysis.

## 5. Conclusions

Serum phosphorus is associated with numerous HF prognostic markers. Elevated sP does not provide independent prognostic information beyond already known markers of syndrome severity. The potential involvement of elevated serum phosphorus as a mediator in important pathways of HF pathophysiology warrants further study.

## 6. Strengths and Limitations

Our study shows the association of high-normal and elevated sP with numerous prognostic HF risk markers in a large homogenous cohort of HF patients with reduced ejection fraction. The lack of dietetic data precluded analysis of nutritional impact on SP. The use of stepwise adjustment for potential confounders pointed out hypothetical mechanisms that lead to elevation of sP in HF. However, the cross-sectional design did not allow analysis of causality.

The lack of measurement of calciotropic and phosphaturic hormones prevents a more in-depth understanding of observed associations.

### 6.1. Clinical Perspectives

In everyday clinical practice in patients with HF, elevated SP cannot be treated as a signature of poor kidney function. It does not have an independent prognostic power, but it may be implicated as a mediator in some aspects of HF pathophysiology.

### 6.2. Translational Look

Future studies should address in more detail the reasons for the elevation of serum phosphorus in CHF as well as the potential direct phosphorus toxicity in cardiovascular system and indirect effects of calciotropic and phosphaturic hormones. The question of potential clinical benefits of interventions that modify phosphorus and/or phosphaturic hormones warrants further studies.

## Figures and Tables

**Figure 1 nutrients-13-04004-f001:**
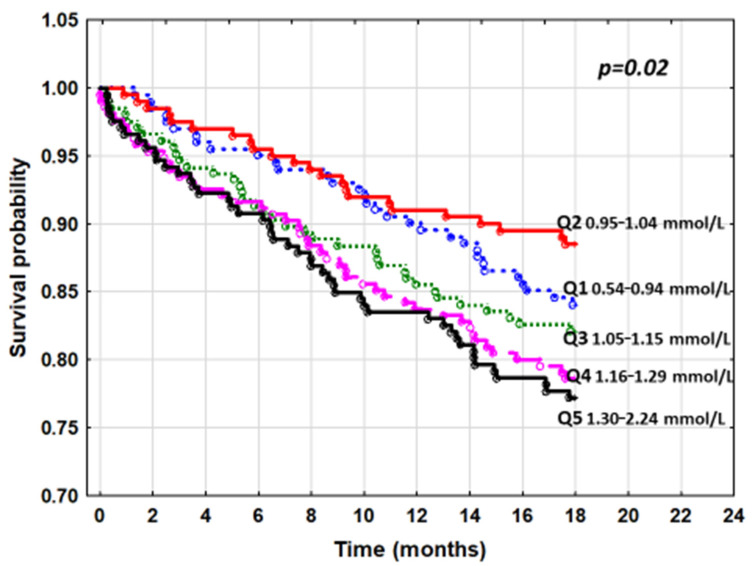
The cumulative survival curves by Kaplan–Meier for quintiles of serum phosphorus and all-cause mortality.

**Figure 2 nutrients-13-04004-f002:**
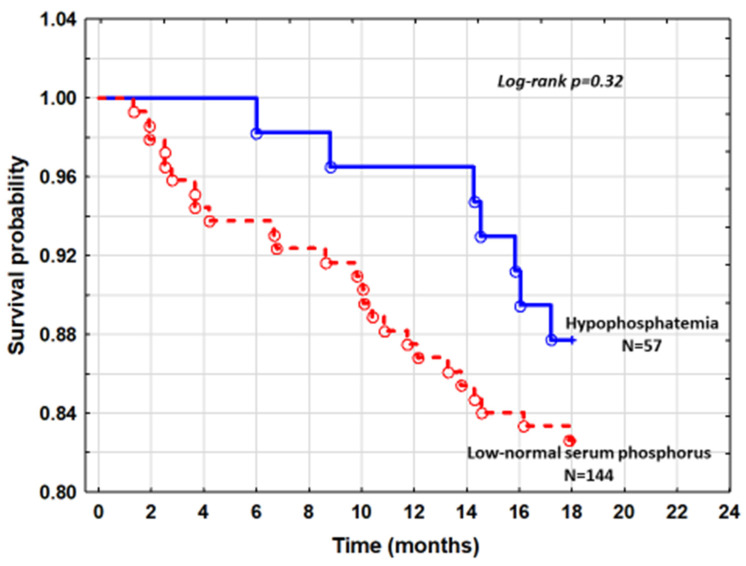
The cumulative survival curves comparing patients with hypophosphatemia and low normal serum phosphorus.

**Figure 3 nutrients-13-04004-f003:**
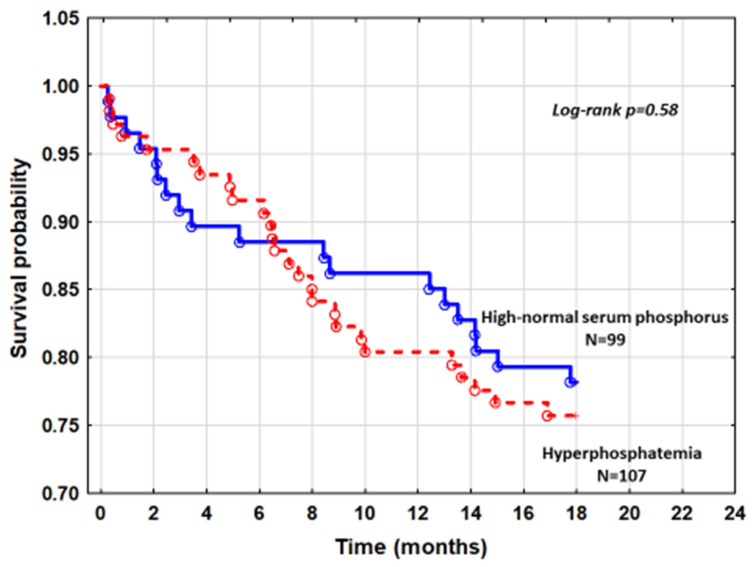
The cumulative survival curves comparing patients with hyperphosphatemia and patients with high-normal SP.

**Figure 4 nutrients-13-04004-f004:**
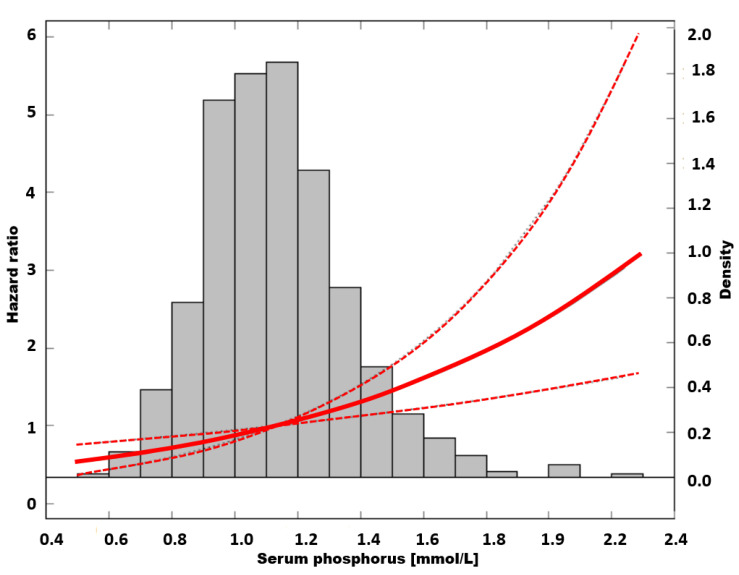
The risk of all-cause mortality in unadjusted model.

**Figure 5 nutrients-13-04004-f005:**
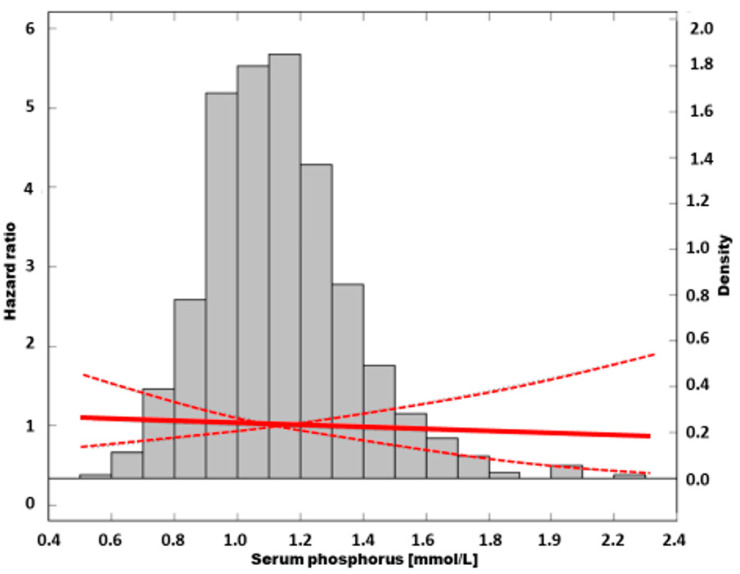
The association of all-cause mortality risk and serum phosphorus in fully adjusted model.

**Figure 6 nutrients-13-04004-f006:**
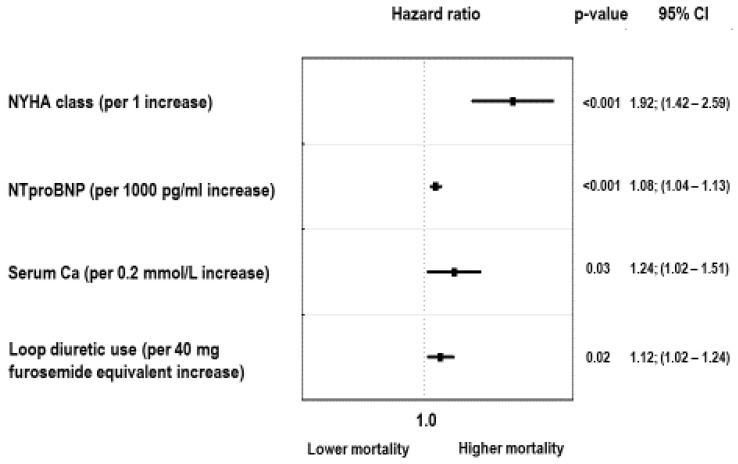
Independent predictors of all-cause mortality in fully adjusted model.

**Figure 7 nutrients-13-04004-f007:**
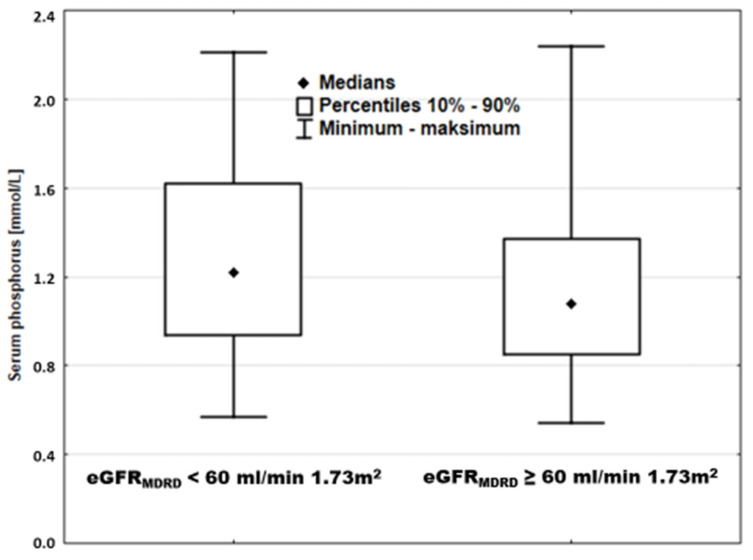
Serum phosphorus in patients stratified according to kidney function.

**Figure 8 nutrients-13-04004-f008:**
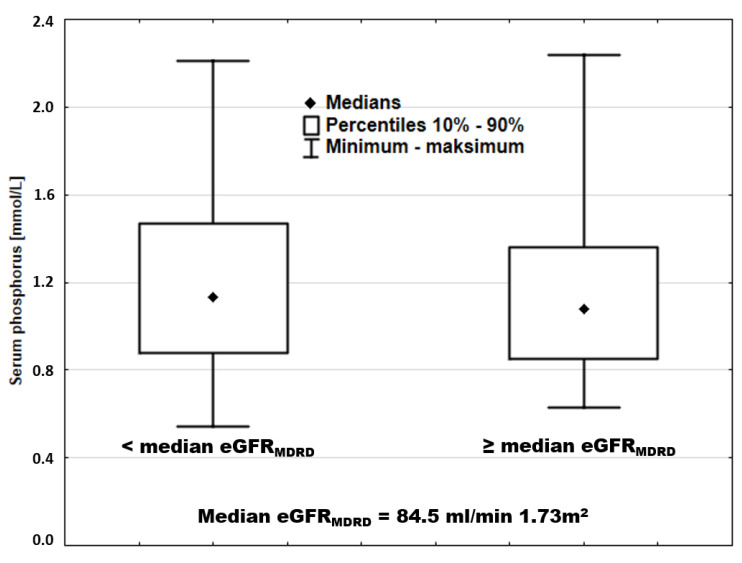
Serum phosphorus in patients stratified according to eGFR_MDRD_.

**Table 1 nutrients-13-04004-t001:** Clinical characteristics of the study cohort. Data are presented as mean ± SD, median value with quartile range, or as percentages.

Feature		Quintiles of sP (mmol/L)					
	All N = 1029	Q1 0.54–0.94, N = 201 (Hypophosphatemia in 57 (28.3%) Patients)	Q2 0.95–1.04 N = 200	Q3 1.05–1.15 N = 207	Q4 1.16–1.29 N = 215	Q5 1.30–2.24, N = 206 (Hyperphosphatemia in 108 (52.4%) Patients)	*p*-Value
Demography							
Sex (% women)	13.7	7.0	10.5	14.6	16.7	18.9	0.002
Age (years)	53 ± 10	54 ± 8	54 ± 10	53 ± 10	52 ± 12	51 ± 12	0.02
BMI (kg/m^2^)	26.4 ± 4.5	27.6 ± 4.1	26.9 ± 4.3	26.0 ± 4.1	25.9 ± 4.6	25.9 ± 4.8	<0.0001
Ischemic etiology (%)	62.9	64.7	68.5	62,6	60.9	58.3	0.26
Duration of HF (months)	35 (56)	40 (54)	37 (44)	32 (42)	38 (56)	28 (56)	0.03
Weight loss in HF (%)	6.7 (13,3)	3.2 (14.6)	5.4 (11.9)	7.8 (13.0)	7.6 (12.7)	9.7 (13.8	<0.0001
Clinical characteristics and echocardiography							
NYHA I (%)	6.3	9.0	8.0	8.5	5.6	1.0	<0.0001
NYHA II (%)	36.5	47.3	40.0	33.5	32.1	30.6	
NYHA III (%)	47.7	40.8	46.0	48.5	50.7	52.4	
NYHA IV (%)	9.5	2.9	6.0	9.5	11.6	16.0	
MVO_2_ (ml/kg*min)	14.6 ± 4.8	14.8 ± 4.1	14.9 ± 4.8	14.9 ± 5.0	14.3 ± 4.8	14.0 ± 4.9	0.12
LVEF (%)	25.2 ± 8	27.5 ± 8	25.9 ± 8	24.8 ± 9	24.4 ± 9	23.3 ± 8	<0.0001
Biochemistry							
eGFR_MDRD_ (mL/min x 1.73 m^2^)	85 (38)	87 (34) (Hypophosphataemic subgroup: 85) (36))	90 (33)	84 (36)	84 (41)	71 (44) (Hyperphosphataemic subgroup: 64) (47))	<0.0001
hsCRP (mg/L)	2.9 (5.6)	2.5 (4.9)	2.6 (4.6)	2.7 (4.4)	3.3 (6.2)	4.5 (7.3)	<0.0001
Sodium (mmol/L)	136 ± 4	137 ± 3	137 ± 3	136 ± 4	136 ± 4	134 ± 4	<0.0001
NTproBNP (pg/mL)	1393 (2538)	1072 (1642)	1083 (366)	1344 (2070)	1917 (3633)	2310 (3151)	<0.0001
Calcium (mmol/L)	2.3 ± 0.18	2.3 ± 0.16	2.3 ± 0.19	2.3 ± 0.17	2.3 ± 0.17	2.4 ± 0.18	<0.0001
Phosphorus (mmol/L)	1.13 ± 0.23	0.84 ± 0.09	0.99 ± 0.03	1.10 ± 0.03	1.22 ± 0.04	1.47 ± 0.17	<0.0001
Comorbidities (N/%)							
Hypertension	55.2	60.2	54.5	58.7	52.1	51.0	0.25
Diabetes mellitus type 2	30.7	25.9	34.0	30.6	28.8	34.5	0.29
Hypercholesterolemia	60.6	60.7	65.0	61.7	60.5	55.3	0.39
History of smoking	72.1	68.2	73.0	71.4	73.5	74.3	0.55
Pharmacotherapy							
ACEI/ARB (%)	93.1	96.5	93.0	91.3	90.2	94.7	0.08
ACEI/ARB (% of recommended dose)	60 ± 51	66 ± 51	68 ± 58	55 ± 40	60 ± 57	52 ± 45	0.002
Beta-blockers (%)	97.6	98.5	99.0	96.6	95.3	98.5	0.07
Beta-blockers (% of recommended dose)	49 ± 30	49 ± 25	51 ± 34	48 ± 31	47 ± 31	51 ± 30	0.34
Aldosterone antagonists (%)	92.3	90.0	94.5	92.2	91.2	93.7	0.43
Aldosterone antagonists (% of recommended dose)	119 ± 65	113 ± 61	108 ± 58	114 ± 58	132 ± 77	128 ± 67	0.001
Loop diuretics (%)	87.2	80.6	87.5	85.0	86.5	96.1	<0.0001
Loop diuretics (mg of furosemide equivalent)	93 ± 82	75 ± 66	77 ± 66	94 ± 87	99 ± 91	119 ± 87	<0.0001
Digoxin (%)	47.5	41.3	43.5	48.1	49.8	54.9	0.05
Al-cause mortality (%)							
At 18 months of follow-up (%)	18.0	15.9 (Hypophosphataemic subgroup: 13.8%)	11.5	17.8	21.4	22.8 (Hyperphosphataemic subgroup: 24.3%)	0.03

Legend: Hypophosphatemia, ≤0.80 mmol/L; hyperphosphatemia, ≥1.40 mmol/L; BMI, body mass index; LVEF, left ventricle ejection fraction; eGFR_MDRD_, estimated glomerular filtration rate (MDRD equation); NTproBNP, N-terminal pro peptide of brain-type natriuretic peptide; ACI/ARB, angiotensin converting enzyme inhibitor/angiotensin II receptor blocker.

**Table 2 nutrients-13-04004-t002:** The risk of all-cause mortality related to increasing levels of sP in unadjusted model and in models adjusted for various clinical and laboratory variables.

	Hazard Ratio, 95% Confidence Intervals, *p*-Value				
	Q1 0.54–0.94	Q2 (Ref.) 0.95–1.04	Q3 1.05–1.15	Q4 1.16–1.29	Q5 1.30–2.24
Unadjusted model	1.40; (0.82–2.39), *p* = 0.22	1.0	1.62; (0.96–2.72), *p* = 0.07	1.98; (1.20–3.26), *p* = 0.008	2.15; (1.30–3.53), *p* = 0.003
Model 1	1.38; (0.80–2.36), *p* = 0.25	1.0	1.60; (0.95–2.70), *p* = 0.08	1.92; (1.16–3.17), *p* = 0.01	1.82; (1.10–3.03), *p* = 0.02
Model 2	1.55; (0.85–2.48), *p* = 0.18	1.0	1.45; (0.86–2.45), *p* = 0.16	1.70; (1.04–2.83), *p* = 0.04	1.76; (1.05–2.95), *p* = 0.03
Model 3	1.26; (0.69–2.30), *p* = 0.45	1.0	0.99; (0.54–1.81), *p* = 0.97	1.23; (0.70–2.18), *p* = 0.47	1.14; (0.64–2.05), *p* = 0.66

Model 1, adjusted for serum calcium; Model 2, adjusted for age, sex, BMI, weight loss, hsCRP, eGFR_MDRD_, LVEF; Model 3, model 2 + ACEI/ARB (yes/no), beta-blockers (yes/no), percent recommended dosages of ACEI/ARB and aldosterone antagonists, loop diuretics use (yes/no), dose of loop diuretics, digoxin (yes/no), NYHA class, serum sodium, NTproBNP.

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
