# Peer review of "Higher Serum Phosphorus Is Not an Independent Risk Factor of Mortality in Heart Failure with Reduced Ejection Fraction"

_nutrients, 2021, doi:10.3390/nu13114004_

Round 1

Reviewer 1 Report

The study tested for the association between serum phosphorus concentration and other known risk factors on mortality in a cohort of monitored patients with heart failure. The number of monitored patients as well as the extent of accompanying data and the methods of evaluation are conclusive and informative. The study is important and another step on the way to understand the implications of increased serum phosphate concentrations and makes it obvious that more research is needed.

Due to the format of the manuscript, especially titles as well as numbers and legends of figures and tables, reading was a bit arduous though. To finalize the review, a clarification is warranted.

Suggestion: It might be interesting to discuss dietetic intake of phosphate shortly as amount and source of ingested phosphates have an effect on serum phosphorus as well as calcium and factors of phosphorus homeostasis even though the fact that an 8h fast was guaranteed diminished the impact.  

Minor points:

Line 24: add “serum” sodium as it is unclear at this point if it is Na intake or serum concentration

Line 23 (after BMI) and line 25 (before “A higher”) delete blank

Line 32: element instead of molecule

Line 34: The deviation of what? Might be intake, excretion, serum concentrations…

Line 43: changes of phosphorus

Line 44: no comma after ‘although’. Association between. Language of “very few studies have shown such…”?

Line 47: delete blank after patients

Line 60: at the date?!

Line 63: exceeding the upper reference range / limit

Line 69: did you determine the highest body weight and the man value for the BW as well?

Line 71: in individual patients

Line 92: the methods are not well described. Were reagents from Roche also used to determine Ca, P, Na? Did you measure total Ca? Why not ionized Ca?

Methods or results – give the reference range for serum P, especially as a footnote for table 1

Line 129: CHF – abbreviation introduced? Chronic heart failure?

Table 1: some parts are bold (all N=1029 etc.), others not (from Q2 onwards)

Line 144: what is the reference range for Ca for the method you used?

Figures: Titles below and above figures? Decimals using comma instead of dots (y-axis)

Figure 2+3: Legend below upper title – different position, not discernable – definition of ‘complete’ and ‘cut’ e.g. in a footnote; title below – where does title end and text start (fig. 3)? Broken y-axis or possible to start with values of 0.7 and 0.8, respectively, and not 0? Title of fig 3 missing? Starts with ‘as a reference…’ = obtuse

Delete blank after survival curves (title fig 2)

Line 170: fig C?

Figure 4 and 5: resolution / quality is poor. Is it ‘Density’. Legend for lines?

Line 198: In fig 2? Maybe 5?

Line 212: in the fully adjusted model is shown in fig – is it 3 or 5?

Line 260: Ess et al.

Line 265: outcome may provide

Figure 24? No title?

Line 285: underscore the high

Line 303: a recent study in rats

Line 304: shown to be a direct

Line 311: In an in-vivo experiment in humans?!

Figure legends – not clear what numbers or titles are correct as they are different (e.g. no 5: the association of all cause mortality risk […] in the text)

Author Response

Dear Reviewer

I thank you very much for your suggestions. All your corrections were inserted into the manuscript. We have improved some sections in order to make them more clear. Also, in obedience to your remarks we have modified figures and corrected their legends. We address comments point by point in red font as follows:

Due to the format of the manuscript, especially titles as well as numbers and legends of figures and tables, reading was a bit arduous though. To finalize the review, a clarification is warranted.

We have done extensive review of the manuscript and some modifications were included in order to make the text more reader friendly

Suggestion: It might be interesting to discuss dietetic intake of phosphate shortly as amount and source of ingested phosphates have an effect on serum phosphorus as well as calcium and factors of phosphorus homeostasis even though the fact that an 8h fast was guaranteed diminished the impact.

Additional comment was added into limitation section

Line 24: add “serum” sodium as it is unclear at this point if it is Na intake or serum concentration

added

Line 23 (after BMI) and line 25 (before “A higher”) delete blank

deleted

Line 32: element instead of molecule

done

Line 34: The deviation of what? Might be intake, excretion, serum concentrations…

corrected

Line 43: changes of phosphorus

corrected

Line 44: no comma after ‘although’. Association between. Language of “very few studies have shown such…”?

deleted and gramma has been improved

Line 47: delete blank after patients

deleted

Line 60: at the date?!

improved

Line 63: exceeding the upper reference range / limit

corrected

Line 69: did you determine the highest body weight and the man value for the BW as well?

The highest body weight is the mean value of measurements available in medical records 

Line 71: in individual patients

corrected

Line 92: the methods are not well described. Were reagents from Roche also used to determine Ca, P, Na? Did you measure total Ca? Why not ionized Ca?

The description was extended, ionized calcium was not measured

Methods or results – give the reference range for serum P, especially as a footnote for table 1

done

Line 129: CHF – abbreviation introduced? Chronic heart failure?

corrected to HF

Table 1: some parts are bold (all N=1029 etc.), others not (from Q2 onwards)

corrected - this error was caused by editing process

Line 144: what is the reference range for Ca for the method you used?

the reference range of total calcium was 2.10 to 2.54 mmol/l.

Figures: Titles below and above figures? Decimals using comma instead of dots (y-axis)

Corrected

Figure 2+3: Legend below upper title – different position, not discernable – definition of ‘complete’ and ‘cut’ e.g. in a footnote; title below – where does title end and text start (fig. 3)? Broken y-axis or possible to start with values of 0.7 and 0.8, respectively, and not 0? Title of fig 3 missing? Starts with ‘as a reference…’ = obtuse

Corrected

Delete blank after survival curves (title fig 2)

deleted

Line 170: fig C?

corrected

Figure 4 and 5: resolution / quality is poor. Is it ‘Density’. Legend for lines?

Line 198: In fig 2? Maybe 5?

Corrected

Line 212: in the fully adjusted model is shown in fig – is it 3 or 5?

corrected

Line 260: Ess et al.

deleted

Line 265: outcome may provide

corrected

Figure 24? No title?

There is no figure 24, corrected

Line 285: underscore the high

corrected

Line 303: a recent study in rats

corrected

Line 304: shown to be a direct

corrected

Line 311: In an in-vivo experiment in humans?!

corrected

Figure legends – not clear what numbers or titles are correct as they are different (e.g. no 5: the association of all cause mortality risk […] in the text)

Corrected

One more time we would like to thank you for all your comments and suggestions.

Reviewer 2 Report

Overall, this manuscript presents interesting information regarding the lack of independent prognostic information from sP in patients with HF. My main and minor comments are as follows:

Main comments: 

-Title implies more investigation on mechanistic role of sP in the pathophysiology, which there was not much data to this effect presented in this study. Suggest changing to more closely reflect the study design and outcomes.

-Could the authors discuss why the treatment was less intense in patients with higher sP? Is there a direction to this relationship that is suspected? Or maybe another correlated factor?

-The paper would benefit from expanded discussion on potential mechanistic links between higher sP and HF - particularly the role of FGF23. 

Minor comments: 

-some grammatical editing needed, particularly in the abstract, but also throughout

-Avoid abbreviations in the abstract

-consider using small "s" in your abbreviation "sP"

-notes in parantheses like "main manuscript" or "online supplement" should be edited out

Author Response

Dear Reviewer,

Thank you very much for all suggestions giving us a chance to improve our manuscript. We address these comments point by point in red font as follows:

Main comments: 

-Title implies more investigation on mechanistic role of sP in the pathophysiology, which there was not much data to this effect presented in this study. Suggest changing to more closely reflect the study design and outcomes.

The title was modified according to your suggestion

-Could the authors discuss why the treatment was less intense in patients with higher sP? Is there a direction to this relationship that is suspected? Or maybe another correlated factor?

In higher quintiles of serum phosphorus the clinical and laboratory profile of HF was more advanced. There may be reciprocal relationship between more advanced profile of HF and less intensive treatment. On one hand, less intensive treatment may cause more advanced HF and higher mortality, on the other hand, due to cross-sectional study design it cannot be excluded that worse tolerance of therapy is a reason of less intensive treatment. 

-The paper would benefit from expanded discussion on potential mechanistic links between higher sP and HF - particularly the role of FGF23. 

A paragraph on potential implications of FGF-23, PTH and vitamin D has been already written in the end of discussion paragraph

 Minor comments: 

-some grammatical editing needed, particularly in the abstract, but also throughout

The manuscript was extensively reviewed and some spelling mistakes corrected

-Avoid abbreviations in the abstract

Removed

-consider using small "s" in your abbreviation "sP"

done

-notes in parantheses like "main manuscript" or "online supplement" should be edited out

removed
